# Improving Feature Learning in Remote Sensing Images Using an Integrated Deep Multi-Scale 3D/2D Convolutional Network

**Haron C. Tinega** [1] , **Enqing Chen** [1,*] **and Divinah O. Nyasaka** [2]

1 School of Electrical and Information Engineering, Zhengzhou University, Zhengzhou 450001, China; tinegaharon@gmail.com
2 Department of Information Communication Technology, Kenya Forest Service, Nairobi 00100, Kenya; dondieki@kenyaforestservice.org
* Correspondence: ieeqchen@zzu.edu.cn; Tel.: +86-371-67781544

**Abstract:** Developing complex hyperspectral image (HSI) sensors that capture high-resolution spatial information and voluminous (hundreds) spectral bands of the earth's surface has made HSI pixel-wise classification a reality. The 3D-CNN has become the preferred HSI pixel-wise classification approach because of its ability to extract discriminative spectral and spatial information while maintaining data integrity. However, HSI datasets are characterized by high nonlinearity, voluminous spectral features, and limited training sample data. Therefore, developing deep HSI classification methods that purely utilize 3D-CNNs in their network structure often results in computationally expensive models prone to overfitting when the model depth increases. In this regard, this paper proposes an integrated deep multi-scale 3D/2D convolutional network block (MiCB) for simultaneous low-level spectral and high-level spatial feature extraction, which can optimally train on limited sample data. The strength of the proposed MiCB model solely lies in the innovative arrangement of convolution layers, giving the network the ability (i) to simultaneously convolve the low-level spectral with high-level spatial features; (ii) to use multiscale kernels to extract abundant contextual information; (iii) to apply residual connections to solve the degradation problem when the model depth increases beyond the threshold; and (iv) to utilize depthwise separable convolutions in its network structure to address the computational cost of the proposed MiCB model. We evaluate the efficacy of our proposed MiCB model using three publicly accessible HSI benchmarking datasets: Salinas Scene (SA), Indian Pines (IP), and the University of Pavia (UP). When trained on small amounts of training sample data, MiCB is better at classifying than the state-of-the-art methods used for comparison. For instance, the MiCB achieves a high overall classification accuracy of 97.35%, 98.29%, and 99.20% when trained on 5% IP, 1% UP, and 1% SA data, respectively.

**Keywords:** convolutional neural networks; deep learning; hyperspectral image classification; integrated networks; multi-scale feature learning; spectral–spatial features; remote sensing

## 1. Introduction

Remote sensing involves the use of sophisticated camera sensors to remotely (from satellite or aircraft) detect and monitor the physical characteristics of a given portion of Earth's surface area using the reflected and emitted radiation [1]. The rapid technological innovation in remote sensing has resulted in the development of complex hyperspectral image (HSI) sensors that capture both voluminous (hundreds) spectral bands and high-resolution spatial information of the earth's surface to produce a three-dimensional (3D) HSI data cube [2,3], as shown in Figure 1.

The original hyperspectral image $H$ is depicted in Figure 1 as a three-dimensional (3D) data cube, with the planes X–Y denoting the spatial data and the Z-axis denoting the spectral bands. Assuming that there are $b$ spectral bands in the original hyperspectral image, then every pixel in $H$ is composed of $b$ spectral bands. Since $H$ is 3D data, the design of

the deep learning-based HSI classifiers shifted to models that can handle both spectral and spatial features and can optimally train on limited HSI sample data [4]. The convolutional neural network (CNN) emerges as the favorite among the deep learning methods proposed for HSI classification because of its ability to extract rich deep features, ensure the integrity of spatial and spectral information, and avoid the initiative and randomness of human feature extraction. As depicted in Figures 2–4, there are three categories of deep CNN feature learning methods based on how HSI features are processed: pre-processing-based, post-processing-based, and integrated-based.

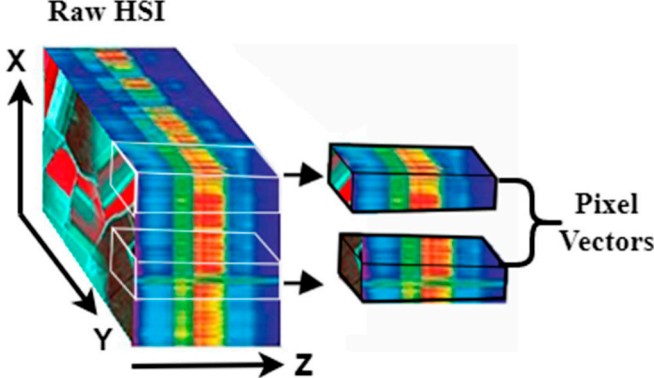

**Figure 1.** The original hyperspectral image data cube and pixel vectors.

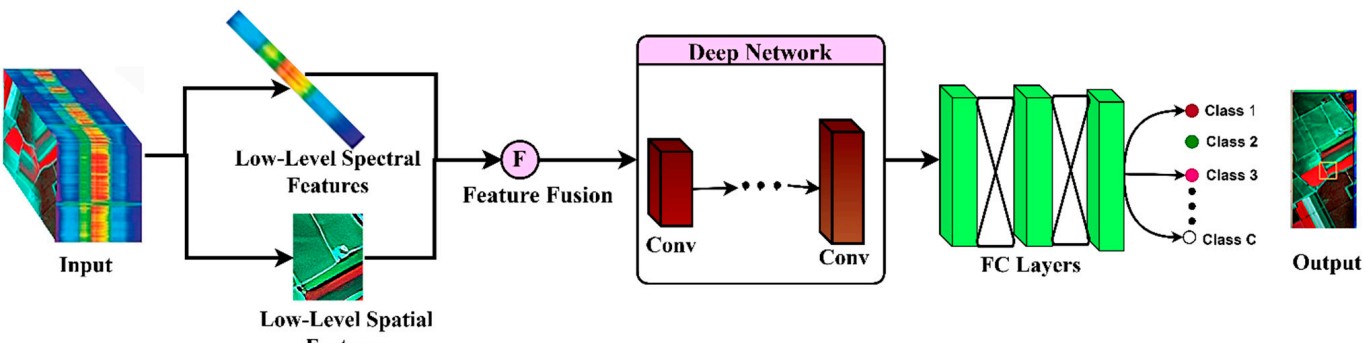

**Figure 2.** The structural design of the pre-processing-based networks.

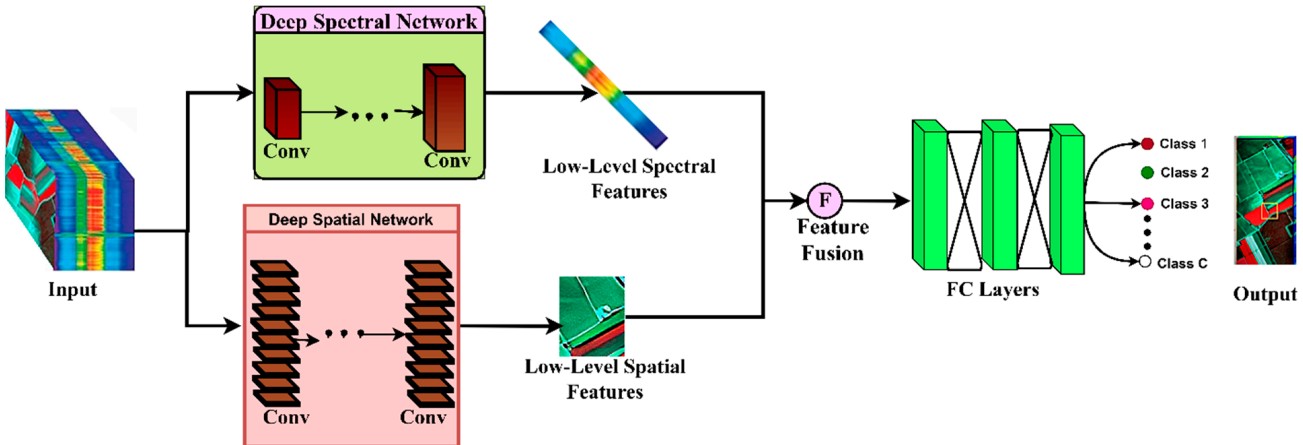

**Figure 3.** The structural design of the post-processing-based networks.

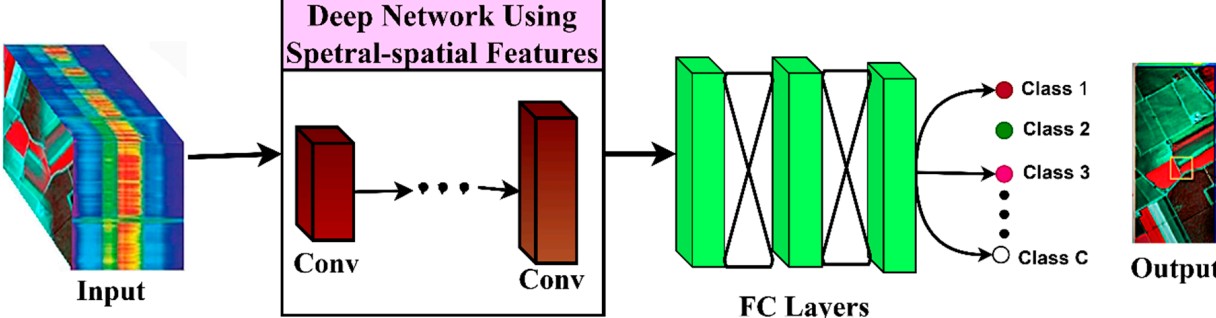

**Figure 4.** The structural design of the integrated networks.

The structural layout of the pre-processing-based methods is depicted in Figure 2. These methods separately acquire the raw HSI data's low-level spectral and spatial features. The extracted low-level spatial data are converted into a 1D input vector before fusing with the original spectral vector. The resultant 1D spectral–spatial feature is channeled into a deep learning network, where further extraction of high-level spectral–spatial features occurs. The resulting highly discriminative spectral–spatial cues are input into the fully connected (FC) layer and subsequently given to the classifier for classification [5,6]. These methods are complex to train as they require high computation memory and large training data.

The structure of the post-processing-based networks is depicted in Figure 3. Using a deep 2D spatial network and a deep 1D spectral network, highly discriminative spatial and spectral properties are extracted, respectively. The output from these two distinct networks is fused before they are passed to the FC layers before being fed into the classifier for classification [7,8]. Hao et al. [2] proposed a novel two-stream deep architecture for HSI classification. However, these methods experience low classification accuracy, especially when subjected to very few training samples.

Unlike the pre-processing and post-processing approaches that exploit the spectral and spatial data separately, the integrated approach shown in Figure 4 simultaneously processes spectral–spatial features directly from the HSI data cube. It often increases their ability to obtain discriminative features robust to nonlinear processing, hence better classification accuracy. Zhong et al. [9], Chen et al. [10], and Li et al. [11] incorporated 3D-CNN layers into their network designs to learn spectral–spatial features from the raw HSI data simultaneously. However, due to the limited training data available in the HSI datasets, utilizing 3D-CNNs alone in developing deep HSI classifiers often results in computationally expensive models prone to overfitting as the network deepens [6,10,12].

Recently, several attempts have been proposed to reduce the dimensionality curse introduced by voluminous spectral bands in HSI data. Among the methods proposed, the principal component analysis (PCA) [13] has gained popularity in hyperspectral imaging [5,6,10,14–16]. Additionally, several methods have been introduced to address the challenge of overfitting in models utilizing an integrated network approach. Some notable integrated approach advancements include the research by Zhong et al. [9], who proposed and designed the SSRN structure with the identity mapping of residual blocks for spectral–spatial feature learning [9]. Lee and Kwon [17] also utilized the residual connections in their design to develop a network that learns hierarchical features [17]. Feng et al. [18] introduced residual connection on HybridSN to develop R-HybridSN, which can optimally train on limited HSI data without overfitting. Other researchers, such as Roy et al. [19], proposed a hybrid model that combines 3D-CNN and 2D-CNN in its network structure to extract spectral–spatial features [19]. Cao and Guo [20] proposed the SSRN, an end-to-end hybrid expansion residual deep convolutional network, which is composed of residual blocks and hybrid dilated convolutions (HDC) [20]. Wu et al. [21] designed the 3D ResNeXt structure using feature fusion and label-smoothing strategies [21]. Tinega et al. [1], developed a GGBN model that used the biological genome concept to combine 3D-CNN and 2D-CNN

and residual connections in a network structure [1]. Among other researchers who utilized a mixture of 3D-CNN and 2D-CNN in their network structure, and residual connections to produce the state-of-the-art models include zhao et al. [22], and Tinega et al. [23].

To further the research on developing deep HSI models that can optimally be trained using limited training samples, we propose an integrated deep multi-scale 3D/2D convolutional network (MiCB). The main contribution of the proposed MiCB model is the creative use of MiCB blocks, which allow the network to convolve low-level spectral features with high-level spatial features that are strengthened by multi-scale kernels, residual connections, and depthwise separable convolutions.

The remainder of this work is structured as follows: Section 2 explains the research technique; Section 3 examines the experimental setup; Section 4 discusses the experimental results and discussion; and Section 5 concludes this research.

## 2. Methodology

### 2.1. The Proposed Model

Figure 5 depicts the whole framework of the suggested MiCB model, which can train optimally using a minimal amount of data. The MiCB network contains three parts: pre-processing, the spectral–spatial feature learning process (MiCB Architecture), and classification.

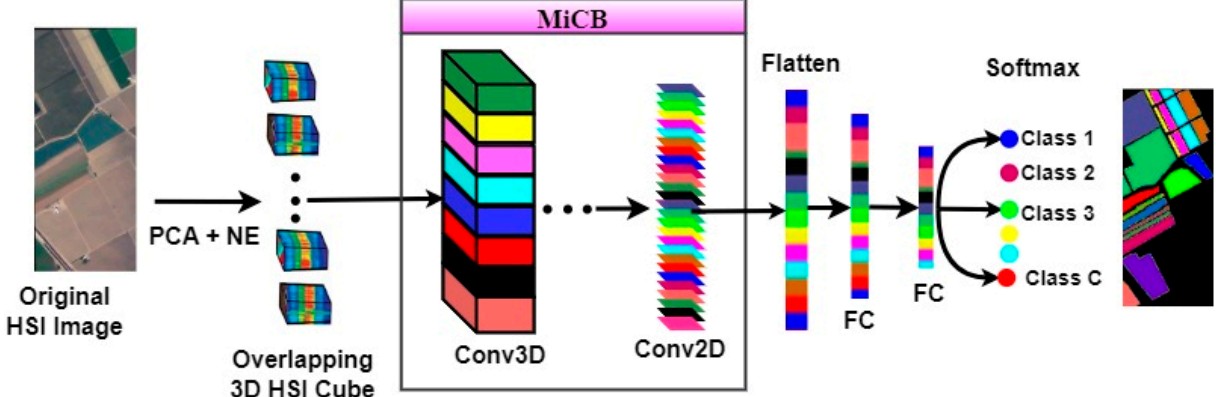

**Figure 5.** Flowchart of the proposed MiCB network.

#### 2.1.1. The Pre-Processing Part

Figure 6 outlines the pre-processing framework of the proposed MiCB network. The dimensionality of an original HSI data cube $H$ can also be stated as $H \in \mathbb{R}^{x \times y \times b}$, where $x$ denotes the width, $y$ denotes the height, and $b$ denotes the number of spectral bands. Due to the presence of the voluminous spectral bands, the first step in pre-processing involves dimensionality reduction in these spectral bands. In this regard, we use the PCA as a dimensionality reduction method. However, the PCA is sensitive to variable variances. Therefore, the first step involves centering and standardizing the HSI data cube $H$ by computing and subtracting the mean value and dividing it by the standard deviation for each spectral band in the original data cube. Mathematically, this can be written as:

$$H = \frac{value - mean}{standard\ deviation} \tag{1}$$

This is followed by the computation of the covariance matrix and the identification of the principal components, which are the data lines with maximal variance. The more significant the variance a line carries, the more information it holds. Therefore, PCA aims to find $t$ number of components $b$ that carry more information without losing valuable information, resulting in the data cube $I \in \mathbb{R}^{x \times y \times t}$, such that $t < b$. The data cube $I \in \mathcal{R}^{x \times y \times t}$ is further subjected to neighborhood extraction (NE), where $G$ overlapping

3D patches of dimensionality $p \times p \times t$ are extracted. The truth label of these overlapping patches at a given spatial location is dependent on the label of the central pixel.

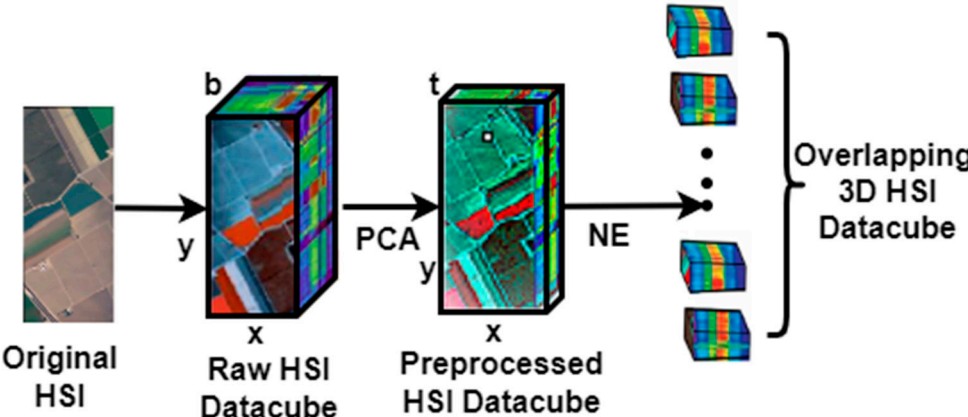

**Figure 6.** HSI data pre-processing framework.

### 2.1.2. Spectral–Spatial Feature Learning Process

Figure 7 shows the detailed architecture of the proposed MiCB, which is based on a hybrid structure of 3D and 2D CNNs. It illustrates how the MiCB model simultaneously convolves low-level spectral cues with high-level spatial features, utilizing mixed 3D/2D CNN layers, multi-scale kernels, depthwise separable convolutions, and residual connections to extract highly discriminative HSI features. Substituting 3D-CNN with 2D-CNN layers increases the network's ability to learn spatial information at the top levels and reduces the model complexity [17]. The usage of non-identity multi-residual connections drastically reduces the challenge of gradient disappearance in the MiCB network [15] while replacing the traditional 2D-CNN with the 2D depthwise separable convolutional layers promotes the reduction in network parameters and prevents overfitting as the model structure deepens. Lastly, the utilization of multi-scale kernels enhances the extraction of abundant contextual features [23–26].

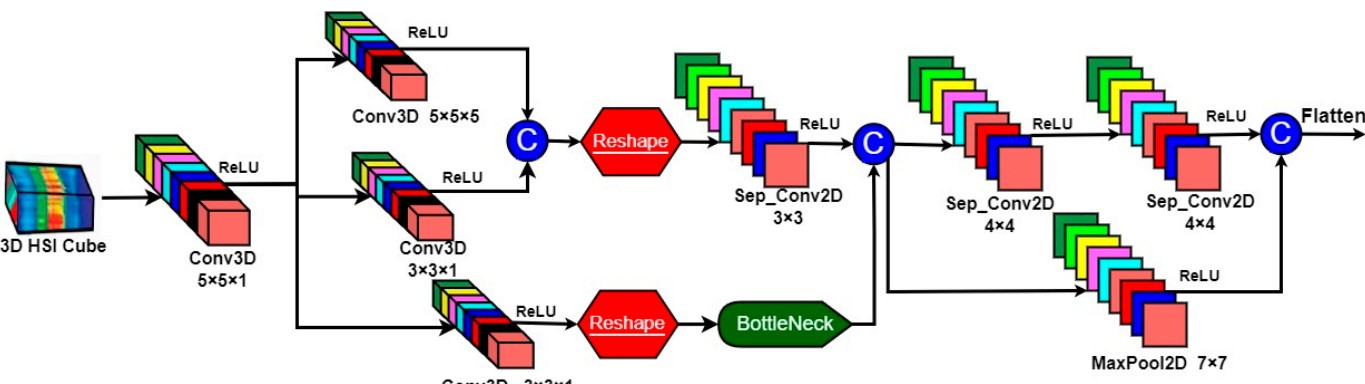

**Figure 7.** The MiCB architecture.

At the bottom of the MiCB network architecture, we utilized the 3D convolution to extract HSI spectral–spatial features, as shown in Figure 8.

A 3D convolution operation, as illustrated in Figure 8, can be denoted as:

$$v_i^{x,y,z} = c_i + \sum_{j=1}^{J} \sum_{r=0}^{R_i-1} \sum_{q=0}^{Q_i-1} \sum_{p=0}^{P_i-1} w_{i,j}^{r,q,p} \times v_{(i-1),j}^{(x+r),(y+q),(z+p)} \tag{2}$$

where $v_i^{x,y,z}$ is the neuron activity at spectral-spatial position $(x, y, z)$, $w_{i,j}^{r,q,p}$ is the Kernel weight of the jth feature map in the *i*th layer at $(r, q, p)$. The $R$, $Q$, and $P$ are the length, width, and depth dimensions of the jth feature map. The $v_{(i-1),j}^{(x+r),(y+q),(z+p)}$, is the 3D convolution output at position $(x, y, z)$ in the jth feature map of the $i-1$ layer. The $c_i$ is the bias value of the convolution filter in the *i*th layer.

We introduced nonlinearity to $v_l^{x,y,z}$ using the rectified linear unit (ReLU) activation function as denoted in Equation (3):

$$R\left(v_l^{x,y,z}\right) = Max\left(0, v_l^{x,y,z}\right) \tag{3}$$

Similarly, a nonlinear 2D convolutional operation in the jth feature map of the ith layer is as shown in Figure 9.

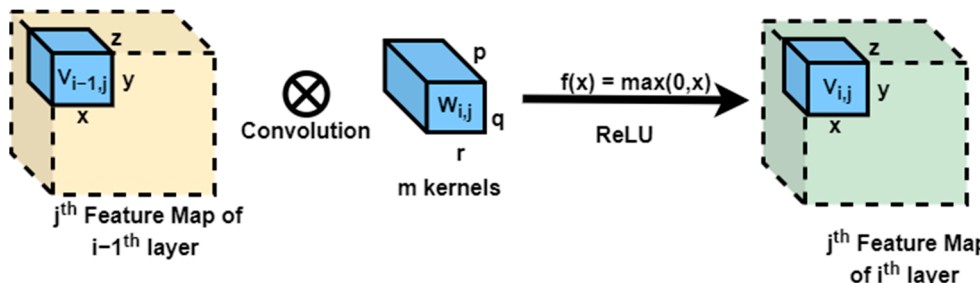

**Figure 8.** A nonlinear 3D convolutional operation in the jth feature map of the ith layer.

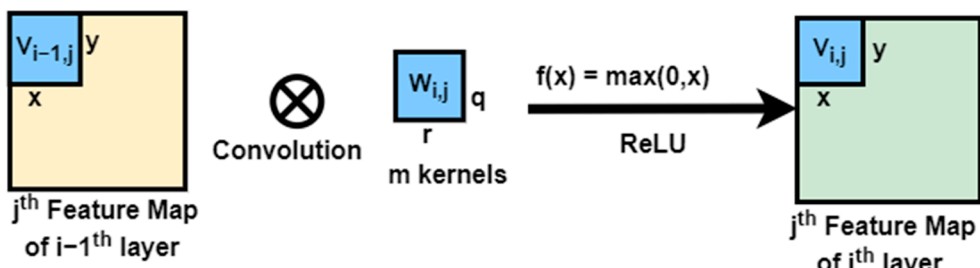

**Figure 9.** A nonlinear 2D convolutional operation at position $(x, y)$ in the jth feature map of the ith layer.

Figure 9 can be expressed mathematically using Equation (4):

$$v_i^{x,y} = R\left(c_i + \sum_{j=1}^{J} \sum_{r=0}^{R_i-1} \sum_{q=0}^{Q_i-1} w_{i,j}^{r,q} \times v_{(i-1),j}^{(x+r),(y+q)}\right) \tag{4}$$

R represents the ReLU activation function.

At the top layer of the proposed MiCB model structure, the 3D feature maps are reshaped into 2D feature maps, as shown in Figure 9.

For example, in Figure 10, the 3D layer of the network has 48 feature maps with a size of $11 \times 11 \times 15$. To learn the feature maps in the 2D space, we reshape the 48 3D feature maps into 720 2D feature maps with dimensions of $11 \times 11$.

We utilized the bottleneck concept shown in Figure 11 to reduce the number of network parameters and mitigate overfitting by downscaling the number of feature maps after reshaping. This is achieved by eliminating redundant features and retaining the smallest possible number of highly discriminative features that preserve the predictive power of the remaining data. In Figure 11, we used a $1 \times 1$ filter to downsample feature maps. The $1 \times 1$ filter has a single weight for each input feature map, making it possible to act like a single neuron with input from the same position across each feature map in the input. Therefore, utilizing a convolutional layer with multiple $1 \times 1$ filters at any point of

the CNN structure allows the depth of the summarized input feature maps to be decreased to increase the network structure's efficiency by reducing the computational costs.

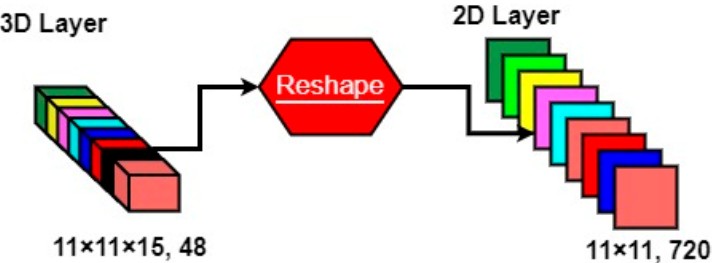

**Figure 10.** Transformation structure of 3D layer to 2D layer.

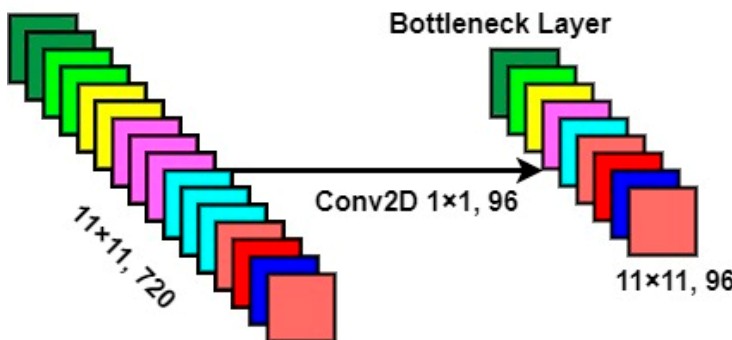

**Figure 11.** Downsampling feature maps using $1 \times 1$ filters.

The MiCB structure integrates feature maps from different convolution layers using a concatenation operation, as shown in Figure 12. Feature concatenation is just a feature-stacking operation. For example, in Figure 12, we have two layers with 125 feature maps each to perform channel-wise feature concatenation; the resulting output will have $125 \times 2 = 250$ concatenated feature maps.

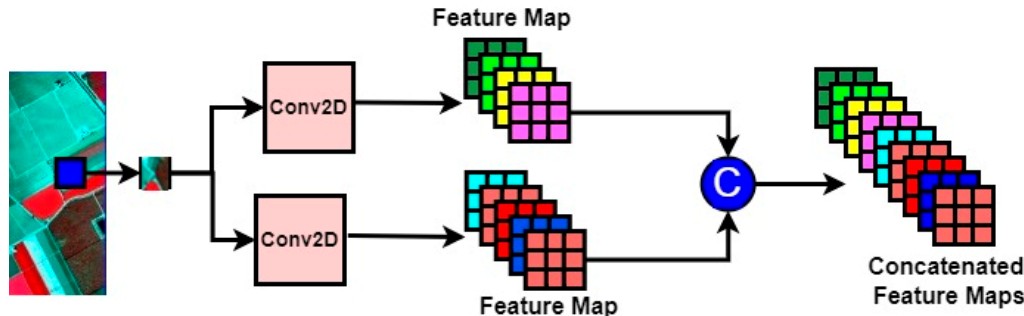

**Figure 12.** Feature concatenation operation.

The 2D feature maps are flattened into a continuous linear vector before being passed into the FC layer. We employ two FC layers to extract deep spectral–spatial features before pushing the features into the classification's softmax layer.

### 2.1.3. MiCB Classifier

The MiCB model employs a softmax layer in the classification part. Softmax is a probabilistic function that measures the relationship between reference and output values. Thus, the chance that a particular input matches a specific class is calculated as follows:

$$\sigma(\vec{t})_i = \frac{e^{t_i}}{\sum_{j=1}^{C} e^{t_j}}$$

$$\text{for } i = 1, \ldots c, \ldots, C, \text{ and } t = t_1, \ldots, t_C \in R^C \tag{5}$$

$\vec{t} = t_1, \ldots, t_C \in R^C$ denotes the input vector values $t_i$, and C denotes the number of classes. The $e^{t_i}$ denotes the standard exponential function, while the $\sum_{j=1}^{C} e^{t_j}$ is the normalization term that ensures that all output values of the standard exponential function sum to 1, and each value should range between 1 and 0 to constitute a proper probability distribution.

Fine-tuning the network is performed using backpropagation. We chose the categorical cross-entropy loss because the number of classes exceeds two. The categorical cross-entropy loss function measures how well our network models the training data. It aims to minimize the training loss between the anticipated and target output, as the lesser the loss, the more accurate the model. Mathematically, the cross-entropy loss can be defined as shown in Equation (6):

$$L = -\sum_{i=1}^{C} r_i \log\left(\sigma(\vec{t})_i\right), \text{ for C classes} \tag{6}$$

$r_i$ is the ground truth label and $\sigma(\vec{t})_i$ is the softmax probability for the ith class.

Finally, the prediction label is decided by taking the argmin value $\hat{t}_l$ of the loss function.

$$\hat{t}_i = \underset{c}{ar\,gmin\,L} \tag{7}$$

The *argmin* operation finds the class with the least loss value from the target function.

### 2.2. Simultaneous Convolution of Low-Level High-Level Spectral–Spatial Features

The main backbone of the MiCB block is the convolution of low-level spectral features with high-level spatial features. This is achieved through the use of 3D kernels that only convolve the spatial dimensions while retaining the spectral aspect. The MiCB model uses kernels of sizes $5 \times 5 \times 1$ and $3 \times 3 \times 1$ to learn the spatial features and preserve low spectral features for convolution at higher network layers (see Figure 7).

### 2.3. Multi-Scale 3D Convolution Block

Figure 13 depicts the framework for multi-scale feature learning with kernels of various sizes to identify a broader range of significant characteristics [23,25–27]. The red, aqua, and blue boxes are distinct convolutional filters used to discover hidden characteristics.

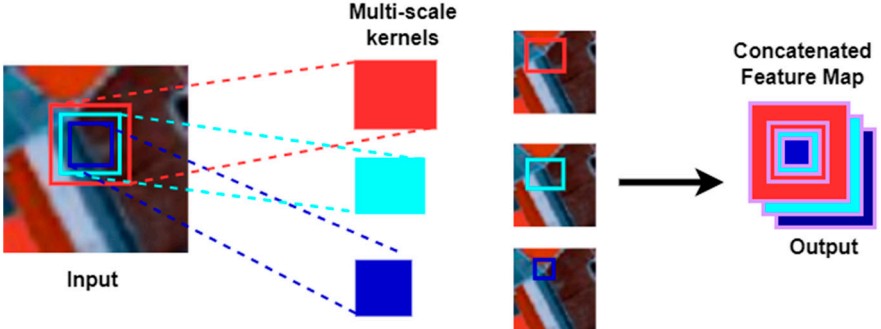

**Figure 13.** The framework of multi-scale feature learning.

### 2.4. Depthwise Separable Convolution

Unlike the conventional 2D convolution that jointly maps spatial and channel information when generating feature maps, the depthwise separable convolution performs

convolution in two steps: the depthwise spatial convolution and the point-wise convolution, as shown in Figure 14.

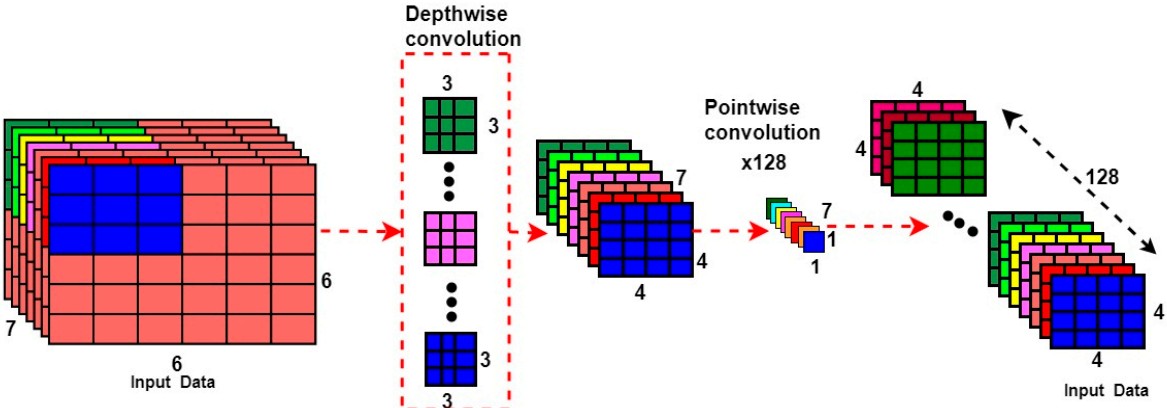

**Figure 14.** Overall depthwise separable convolution operation.

The depthwise spatial convolution performs the 2D convolution on each feature map separately. As illustrated in Figure 14, the input data is comprised of 7 feature maps of size 6×6. We use seven kernels, separately of size $3 \times 3$. Each kernel convolves only one channel of the input layer resulting in a feature map of dimension $4 \times 4$. We assume that padding is zero and the stride value is set to one. We then stack the resultant seven feature maps of dimensionality $4 \times 4$ to create seven images of size $4 \times 4$. In the second step, we apply point-wise convolution, we extend the depth of the image by applying the $1 \times 1$ convolution with the kernel size of $1 \times 17$ will result in the feature map of dimension $4 \times 4$. Hence after applying $1 \times 1128$ convolutions, we have the final output layer with dimension $4 \times 4128$, as shown in Figure 14. Therefore, we transform an input data of dimension 6 $\times 67$ into an output layer with dimension $4 \times 4128$. The depthwise separable convolution reduces the model complexity regarding network parameters and the calculation time. From the illustration on Figure 14 the method cuts down the parameters from 733, 824 ($3 \times 3 \times 7 \times 16 \times 128$) to 15, 344 ($3 \times 3 \times 7 \times 16 + 1 \times 1 \times 7 \times 16 \times 128$). However, its ability to reduce calculation time is illustrated in Section 4. Hyperspectral image classification has utilized these benefits, which help with information balancing for layers with depth dimensions far larger than spatial dimensions.

*2.5. Residual Learning*

Deep learning research indicates that the depth of the network is more advantageous than its width [9,28]. Deeper networks can learn highly discriminative features; however, training them on limited sample data often results in overfitting [23]. To address the challenge of overfitting, we applied residual connections to sufficiently recover the lost features in the MiCB model as the network depth increases [12]. The architecture of the proposed MiCB network employs non-identical residual connections, shown in Figure 15a,b in its design.

The first non-identical residual connection shown in Figure 15a is a 3D-CNN layer that only convolves spatial dimension + ReLU, utilized at the bottom part of the MiCB network structure among the successful 3D/2D CNN layers to conduct dimension adjustment and facilitate low-level spectral and high-level spatial convolution. We used a convolution layer + ReLU because it converts a network into layers of directed acyclic graphs in which each branch can independently learn highly discriminative features. In deep learning models, feature degradation is at its peak at the top part of the model. In order to recover lost features and control overfitting at this point, we utilized a max pooling layer (shown in Figure 15b) to conduct dimension adjustment. The max pooling function divides the

input feature map into smaller regions and outputs the maximum value from each region. Mathematically, this can be expressed as:

$$g_i^j = p_{max}\left(h_i^j\right) \tag{8}$$

$g_i^j$ denotes the pooled feature map, and $p_{max}(.)$ is a max pooling function.

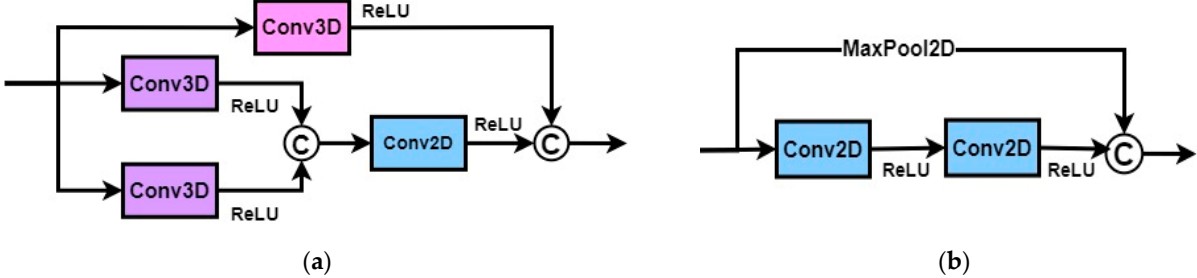

(**a**)                                    (**b**)

**Figure 15.** Non-identical residual connection using (**a**) a 3D-CNN layer + ReLU for dimension adjustment, and (**b**) a max pooling layer for dimension adjustment.

## 3. Experimental Setup

This section contains the dataset description, comparison methods, implementation details, evaluation indicators, experimental results, and discussion.

### 3.1. Dataset Description

Three publicly accessible HSI datasets, IP, UP, and SA, were used to evaluate the effectiveness of the MiCB method. The original IP dataset image dimensions are 145 × 145 × 224. However, due to water absorption, 24 bands were discarded [19,29,30]. Therefore, this experiment uses an IP image dataset with dimensions of 145 × 145 × 200. Its ground truth consists of sixteen classes that are not mutually exclusive, as shown in Table 1 [26]. The IP dataset is the most unbalanced, followed by the UP and SA datasets (see Table 1). The original UP dataset image dimensions are 610 × 340 × 115. We reduced the spectral dimensions from 115 to 103 bands by removing 12 noisy bands [19,29,30]. Its ground truth contains nine classes, as shown in Table 1. The size of the original SA image is 512 × 217 × 204 pixels after discarding 20 water absorption bands. Nevertheless, we anticipated that the results obtained in the training SA dataset, even at less than 1%, would be superior to those of IP and UP, as the majority of classes are adequately represented even when trained using 1% of the data sample. The SA dataset has 16 class labels assigned to the land cover [26].

**Table 1.** The IP, UP, and SA datasets, class labels, and per class sample data percentage.

| Class No | IP Dataset | | SA Dataset | | UP Dataset | |
|---|---|---|---|---|---|---|
| | Class Label | Samples (%) | Class Label | Samples (%) | Class Label | Samples (%) |
| 1 | Alfalfa | 0.45 | Brocoli_green_weeds_1 | 3.71 | Asphalt | 15.50 |
| 2 | Corn-notill | 13.93 | Brocoli_green_weeds_2 | 6.88 | Meadows | 43.60 |
| 3 | Corn-mintill | 8.10 | Fallow | 3.65 | Gravel | 4.91 |
| 4 | Corn | 2.31 | Fallow_rough_plow | 2.58 | Trees | 7.16 |
| 5 | Grass-pasture | 4.71 | Fallow_smooth | 4.95 | Painted | 3.14 |
| 6 | Grass-trees | 7.12 | Stubble | 7.31 | Bare | 11.76 |
| 7 | Grass-pasture-mowed | 0.27 | Celery | 6.61 | Bitumen | 3.11 |
| 8 | Hay-windrowed | 4.66 | Grapes_untrained | 20.82 | Self-Blocking | 8.61 |
| 9 | Oats | 0.20 | Soil_vinyard_develop | 11.46 | Shadows | 2.21 |
| 10 | Soybean-notill | 9.48 | Corn_senesced_green_weeds | 6.06 | | |
| 11 | Soybean-mintill | 23.95 | Lettuce_romaine_4wk | 1.97 | | |
| 12 | Soybean-clean | 5.79 | Lettuce_romaine_5wk | 3.56 | | |
| 13 | Wheat | 2.00 | Lettuce_romaine_6wk | 1.69 | | |
| 14 | Woods | 12.34 | Lettuce_romaine_7wk | 1.98 | | |
| 15 | Buildings-Grass-Trees-Drives | 3.77 | Vinyard_untrained | 13.43 | | |
| 16 | Stone-Steel-Towers | 0.91 | Vinyard_vertical_trellis | 3.34 | | |

### 3.2. Implementation Details

All the tests in this paper were performed online using Google Colab Inc., Mountain View, CA, USA. We used random subsampling in MiCB, which randomly selects some data as the training data while the remaining data are used for testing the model. The results are reported as the mean of seven separate tests. A grid search was utilized to determine the optimal optimizer, learning rate, batch size, epochs, and dropout technique for the MiCB model. For both the IP and UP datasets, the Adam optimizer with a learning rate of 0.0009 and a dropout of 0.6 was selected. However, we adjusted the learning rate for SA to 0.0007 but retained the 0.6 dropout. The number of epochs was set to 100 for all datasets, and the batch size was set to 64, 128, and 96 for the IP, UP, and SA, respectively. To evaluate objectively, we extracted the same spatial–spectral dimension of the overlapping 3D patches of size $17 \times 17 \times 15$ across all datasets (see Tables 2–4).

**Table 2.** The MiCB model's performance on different window sizes when trained using 5% of the IP data sample and assessed in terms of Kappa, OA, and AA.

| Evaluation | 15 ×15 | 17 × 17 | 19 × 19 | 21 × 21 | 23 × 23 | 25 × 25 | 27 × 27 |
|:---:|:---:|:---:|:---:|:---:|:---:|:---:|:---:|
| Kappa | 0.967 | 0.970 | 0.967 | 0.965 | 0.965 | 0.963 | 0.963 |
| OA | 97.14 | 97.35 | 97.13 | 96.90 | 96.95 | 96.76 | 96.71 |
| AA | 90.08 | 92.16 | 91.38 | 89.18 | 91.46 | 91.22 | 91.79 |

**Table 3.** The MiCB model's performance on different window sizes when trained using 1% of the UP data sample and assessed in terms of Kappa, OA, and AA.

| Evaluation | 15 × 15 | 17 × 17 | 19 × 19 | 21 × 21 | 23 × 23 | 25 × 25 | 27 × 27 |
|:---:|:---:|:---:|:---:|:---:|:---:|:---:|:---:|
| Kappa | 0.974 | 0.977 | 0.973 | 0.974 | 0.970 | 0.969 | 0.969 |
| OA | 98.03 | 98.23 | 97.96 | 98.07 | 97.73 | 97.66 | 97.66 |
| AA | 96.44 | 96.73 | 95.99 | 96.09 | 95.51 | 94.98 | 94.99 |

**Table 4.** The MiCB model's performance on different window sizes when trained using 1% of the SA data sample and assessed in terms of Kappa, OA, and AA.

| Evaluation | 15 × 15 | 17 × 17 | 19 × 19 | 21 × 21 | 23 × 23 | 25 × 25 | 27 × 27 |
|:---:|:---:|:---:|:---:|:---:|:---:|:---:|:---:|
| Kappa | 0.988 | 0.991 | 0.992 | 0.995 | 0.995 | 0.995 | 0.996 |
| OA | 98.93 | 99.20 | 99.28 | 99.51 | 99.54 | 99.56 | 99.64 |
| AA | 98.95 | 99.13 | 99.16 | 99.42 | 99.35 | 99.47 | 99.51 |

### 3.3. Evaluation Criteria

We evaluated the performance of the recommended HSI models using the Kappa Coefficient (Kappa), Overall Accuracy (OA), and Average Accuracy (AA) metrics. The OA calculates the percentage of correctly categorized samples. These are samples whose projected outcomes exactly matched the ground truth label. The AA calculates the mean of per-class accuracies. The Kappa coefficient's values range between 0 and 1; a 0 value indicates no consistency while a 1 value indicates a perfect consistency between the classification map and its corresponding ground truth.

## 4. Experimental Results and Discussion

This section presents the experimental results and discussion of the proposed MiCB model in contrast to various cutting-edge approaches using the three (IP, UP, and SA) datasets, including M3D-CNN [27], SSRN [9], R-HybridSN [16], HybridSN [18], and GGBN [1].

*4.1. Effect of Varying Window Size*

This section examines the impact of altering the window size of the MiCB model across the three (IP, UP, and SA) datasets. We utilized a 1% training set for SA and UP and a 5% training set for IP.

Tables 2–4 shows that a spatial window size of $17 \times 17$ attained better classification accuracy on IP and UP datasets, while a $27 \times 27$ spatial window was optimal on the SA dataset. However, we choose a spatial window size of $17 \times 17$ across the three datasets for a fair comparison.

*4.2. Ablation Results*

Models A and B were developed to determine the impact of residual learning and depthwise separable convolution on MiCB classification results. Model A utilizes residual connections with a mixture of the traditional 2D- and 3D-CNN layers. On the other hand, Model B replaced the traditional 2D-CNN layers with depthwise separable convolutions but lacked residual connections. The classification outcomes are displayed in Tables 5–7.

**Table 5.** The summary of the results of the per-class accuracy, Kappa, OA, and AA of Model A, Model B, MiCB, and other selected models trained using 5% of the IP data sample.

| Class Number | Overall Accuracy in Percentage (%) | | | | | | | |
|---|---|---|---|---|---|---|---|---|
| | M3D-CNN | HybridSN | R-HybridSN | SSRN | GGBN | Model A | Model B | MiCB |
| 1 | 27.5 | 61.82 | 45 | 12.99 | 46.36 | 59.09 | 32.14 | 72.73 |
| 2 | 59.15 | 92.25 | 95.45 | 93.04 | 94.78 | 95.15 | 94.85 | 96.05 |
| 3 | 45.07 | 92.97 | 97.36 | 93.72 | 98.38 | 97.97 | 98.70 | 99.24 |
| 4 | 38.49 | 78.22 | 94.8 | 72.38 | 94.49 | 93.27 | 88.57 | 92.38 |
| 5 | 70.33 | 96.6 | 98.85 | 98.16 | 99.15 | 99.69 | 99.38 | 98.79 |
| 6 | 97.2 | 98.11 | 99.32 | 99.86 | 98.02 | 99.55 | 98.66 | 98.52 |
| 7 | 18.52 | 68.52 | 95.56 | 0 | 87.78 | 77.25 | 12.70 | 98.41 |
| 8 | 98.04 | 99.96 | 100 | 99.94 | 99.85 | 99.81 | 99.31 | 99.94 |
| 9 | 25.79 | 83.68 | 65.26 | 0 | 78.95 | 82.71 | 10.53 | 50.38 |
| 10 | 55.85 | 96.12 | 95.9 | 91.01 | 97.82 | 97.04 | 96.36 | 97.26 |
| 11 | 76.2 | 96.66 | 98.09 | 95.63 | 97.98 | 97.86 | 98.21 | 98.73 |
| 12 | 33.89 | 85.44 | 89.15 | 87.9 | 92.97 | 93.45 | 92.01 | 93.12 |
| 13 | 91.23 | 94.97 | 99.74 | 98.53 | 97.64 | 99.71 | 99.49 | 99.34 |
| 14 | 94.68 | 99.34 | 99.26 | 99.82 | 99.03 | 99.97 | 99.14 | 99.77 |
| 15 | 42.37 | 82.92 | 87.66 | 82.09 | 92.29 | 92.45 | 86.22 | 89.96 |
| 16 | 49.32 | 80 | 88.18 | 82.31 | 88.75 | 82.79 | 89.94 | 89.94 |
| Kappa | 0.642 | 0.934 | 0.96 | 0.923 | 0.96 | 0.965 | 0.955 | 0.97 |
| OA (%) | 68.88 | 94.24 | 96.46 | 93.39 | 96.85 | 96.92 | 96.07 | 97.35 |
| AA (%) | 57.73 | 87.97 | 90.6 | 75.28 | 91.51 | 91.67 | 81.01 | 92.16 |

**Table 6.** The summary of the results of the per-class accuracy, Kappa, OA, and AA of Model A, Model B, MiCB, and other selected models trained using 1% of the UP data sample.

| Class | Overall Accuracy in Percentage (%) | | | | | | | |
|---|---|---|---|---|---|---|---|---|
| | M3D-CNN | HybridSN | R-HybridSN | SSRN | GGBN | Model A | Model B | MiCB |
| 1 | 90.56 | 95.72 | 96.94 | 98.76 | 98.50 | 97.65 | 98.56 | 99.40 |
| 2 | 89.47 | 99.68 | 99.69 | 99.91 | 99.70 | 99.68 | 99.77 | 99.38 |
| 3 | 59.11 | 84.38 | 87.17 | 85.72 | 89.03 | 95.37 | 89.70 | 95.01 |
| 4 | 93.25 | 87.7 | 89.15 | 94.85 | 93.28 | 92.57 | 93.26 | 93.90 |
| 5 | 93.66 | 98.99 | 99.51 | 99.76 | 99.71 | 98.83 | 99.77 | 99.47 |
| 6 | 69.63 | 96.82 | 98.44 | 96.11 | 99.79 | 99.26 | 97.54 | 99.73 |
| 7 | 65.71 | 84.42 | 95.82 | 95.98 | 98.14 | 97.13 | 84.40 | 93.97 |
| 8 | 78.35 | 89.18 | 93.28 | 94.96 | 96.03 | 97.93 | 91.50 | 96.89 |
| 9 | 94.41 | 71.71 | 77.82 | 99.89 | 97.35 | 97.77 | 92.42 | 92.79 |
| Kappa | 0.798 | 0.935 | 0.955 | 0.97 | 0.975 | 0.977 | 0.960 | 0.977 |
| OA (%) | 84.63 | 95.09 | 96.59 | 97.67 | 98.13 | 98.30 | 97.01 | 98.29 |
| AA (%) | 81.57 | 89.84 | 93.09 | 96.22 | 96.84 | 97.35 | 94.10 | 96.73 |

**Table 7.** The summary of the results of the per-class accuracy, Kappa, OA, and AA of Model A, Model B, MiCB, and other selected models trained using 1% of the SA data sample.

| Class | Overall Accuracy in Percentage (%) | | | | | | | |
|---|---|---|---|---|---|---|---|---|
| | **M3D-CNN** | **HybridSN** | **R-HybridSN** | **SSRN** | **GGBN** | **Model A** | **Model B** | **MiCB** |
| 1 | 94.88 | 99.99 | 100 | 100 | 99.95 | 100.00 | 99.98 | 99.99 |
| 2 | 99.61 | 100 | 99.97 | 100 | 100 | 100.00 | 99.97 | 100 |
| 3 | 91.89 | 99.82 | 99.49 | 99.96 | 99.92 | 99.74 | 100.00 | 99.72 |
| 4 | 98.33 | 98.38 | 98.72 | 99.72 | 96.25 | 99.68 | 99.49 | 99.66 |
| 5 | 98.83 | 99.26 | 98.43 | 98.73 | 99.33 | 99.34 | 99.60 | 99.02 |
| 6 | 98.09 | 99.93 | 99.9 | 100 | 99.92 | 99.98 | 99.65 | 99.80 |
| 7 | 97.67 | 99.95 | 99.96 | 99.99 | 99.98 | 99.98 | 100.00 | 99.85 |
| 8 | 82.4 | 97.77 | 98.23 | 95.06 | 99.25 | 99.38 | 98.60 | 98.98 |
| 9 | 98.14 | 99.99 | 99.99 | 100 | 100 | 99.99 | 100 | 100 |
| 10 | 87.6 | 98.36 | 97.9 | 98.33 | 99.08 | 98.48 | 98.49 | 98.54 |
| 11 | 86.72 | 96.06 | 96.46 | 97.42 | 98.75 | 96.49 | 96.85 | 99.24 |
| 12 | 96.99 | 97.44 | 99.09 | 100 | 99.77 | 99.92 | 99.48 | 99.48 |
| 13 | 97.14 | 97.42 | 82.82 | 93.02 | 93.67 | 96.79 | 96.35 | 95.86 |
| 14 | 91.78 | 99.52 | 97.25 | 95.62 | 99.27 | 99.10 | 99.30 | 98.00 |
| 15 | 64.42 | 97.06 | 95.12 | 88.18 | 98.49 | 95.76 | 95.18 | 98.02 |
| 16 | 78.14 | 100 | 99.71 | 99.49 | 99.99 | 99.88 | 99.52 | 99.98 |
| Kappa | 0.867 | 0.985 | 0.98 | 0.966 | 0.992 | 0.989 | 0.986 | 0.991 |
| OA (%) | 88.02 | 98.72 | 98.25 | 96.94 | 99.29 | 99.00 | 98.74 | 99.20 |
| AA (%) | 91.41 | 98.81 | 97.69 | 97.84 | 98.98 | 99.03 | 98.90 | 99.13 |

4.2.1. The Summary of Classification Accuracies of the Selected Models Trained Using Very Minimal Sample Data

In this subsection, we present the summary result of the per-class accuracy, Kappa, OA, and AA of Model A, Model B, MiCB, and other selected models such as M3D-CNN, HybridSN, R-HybridSN, SSRN, and GGBN trained on 5% of the IP dataset's total sample data and 1% of the UP and SA datasets' total sample data.

The proposed MiCB outperforms the M3D-CNN, HybridSN, R-HybridSN, and SSRN models on the three (IP, UP, and SA) datasets, as shown in Tables 5–7. However, its performance is comparable with the GGBN on all three datasets. For instance, over the IP dataset, as shown in Table 5, the MiCB model increased the overall classification accuracy of M3D-CNN, HybridSN, R-HybridSN, and SSRN by +28.47%, +3.11%, +0.89%, and +3.96%, respectively. However, the proposed MiCB slightly increased the performance of the GGBN by +0.5%. A similar trend is observed for the UP and SA datasets, where the MiCB increased the overall classification accuracy of M3D-CNN, HybridSN, R-HybridSN, and SSRN by +13.66%, +3.2%, +1.7%, +0.62% on UP (see Table 6) and by +11.18%, +0.48%, +0.95%, +2.26% on SA (see Table 7), respectively. Similar to the MiCB performance on IP dataset, the proposed model slightly increased the overall classification accuracy of GGBN by +0.16% on UP and comparable on the SA dataset.

The M3D-CNN achieved the lowest classification accuracies due to its structural nature, which primarily extracts multi-scale spectral information and insufficient spectral–spatial features. In addition, the M3D-CNN lacks residual connections to recover lost features and prevent overfitting in deep networks when training samples are very few. The SSRN method, on the other hand, recorded better classification accuracies than M3D-CNN across all the tested datasets because it introduced skip connections in its network structure, which prevents degradation. The HybridSN utilized 3D and 2D CNN layers in its network structure to extract highly discriminative HSI characteristics. However, the overfitting problem worsened as the amount of training data declined. The R-HybridSN addressed the HybridSN's limitations by introducing residual connection and depthwise separable convolutions in its network structure to achieve higher classification accuracy. In order to improve the classification accuracies when training on very little sample data, the GGBN incorporated the biological genome approach to judiciously utilize the 3D-CNN

and 2D-CNN layers in its network structure to achieve comparable classification accuracies with the proposed MiCB model.

The high performance of the proposed MiCB model in HSI classification can be attributed to the utilization of residual connection, depthwise separable layers, multi-scale kernels for feature extraction, and its ability to convolve the low-level spectral with high-level spatial features. The effect of residual connection can be observed in classes with very few training samples. For example, the models with residual connections, such as Model A and MiCB, record very high classification accuracies in classes with extremely few training sample data compared to Model B which lacks residual connection in its network structure (See Table 5, class 7 and 9). On the IP dataset, the proposed MiCB model improved the OA and AA of Model A by +0.43% and +0.40%, respectively, and when compared with Model B, an increase of +1.28% in OA and +11.06% of AA was recorded. A similar trend was recorded in the SA dataset (see Table 7), where the MiCB model increased the OA and AA of Model A by +0.20% and +0.10% and Model B by +0.46% and +0.23%, respectively. In the UP dataset (see Table 6), the MiCB model recorded an increase in AO and AA for Model B; however, there was a decrease in OA and AA for Model A.

### 4.2.2. Computational Complexity of Model A, Model B, and MiCB over IP, UP, and SA Datasets

This section illustrates the computational complexity of the proposed MiCB model and its variants in terms of the network parameters and testing time.

We can observe in Table 8 that Model B, with no residual connection, recorded the lowest number of network parameters and the shortest test time length. A similar observation is made when replacing the traditional 2D layers in Model A with depthwise separable layers. Since the MiCB model is a hybrid of Model A and B, adding residual connection increases the computational cost, while replacing the traditional 2D layers with depthwise separable layers leads to reduced computational cost. Hence, the proposed model exhibits balanced network parameters and test time length.

**Table 8.** The network parameters and testing time in seconds over IP, UP, and SA datasets for Model A, B, and MiCB.

| Dataset | Model A | | | Model B | | | MiCB | | |
|---------|---------|------------|-----------|---------|-----------|-----------|---------|-----------|-----------|
| | Params | Train Time | Test Time | Params | Test Time | Test Time | Params | Test Time | Test Time |
| IP | 2,354,700 | 43.01 | 2.95 | 426,108 | 35.08 | 2.07 | 958,428 | 39.96 | 2.47 |
| UP | 2,353,797 | 41.88 | 11.88 | 425,205 | 33.24 | 8.99 | 957,525 | 42.02 | 10.82 |
| SA | 2,354,700 | 47.83 | 14.78 | 426,108 | 40.92 | 11.08 | 958,428 | 45.44 | 12.79 |

For instance, Model A has 1,396,272 more parameters and increased the test time across all datasets by +0.48, +1.06, and +1.99 s, respectively, compared with the MiCB model. However, Model B recorded 532,320 fewer trainable parameters across all datasets than the proposed MiCB model. In terms of the testing time, Model B recorded −0.40, −1.83, and −1.71 s less than the MiCB model across all three datasets, respectively. These observations illustrate the effect of residual and depthwise separable layers on model complexity.

### 4.3. The Training Accuracy and Loss Convergence Graphs

This subsection uses the training accuracy and loss convergence graphs to demonstrate the competitiveness of the proposed MiCB model in comparison with the selected state-of-the-art models over the IP, UP, and SA datasets. Figure 16a–c shows that the proposed MiCB model converges faster than all the compared methods on the IP dataset and is commensurate with the R-HybridSN and HybridSN methods on SA and UP datasets but faster than the SSRN and GGBN models.

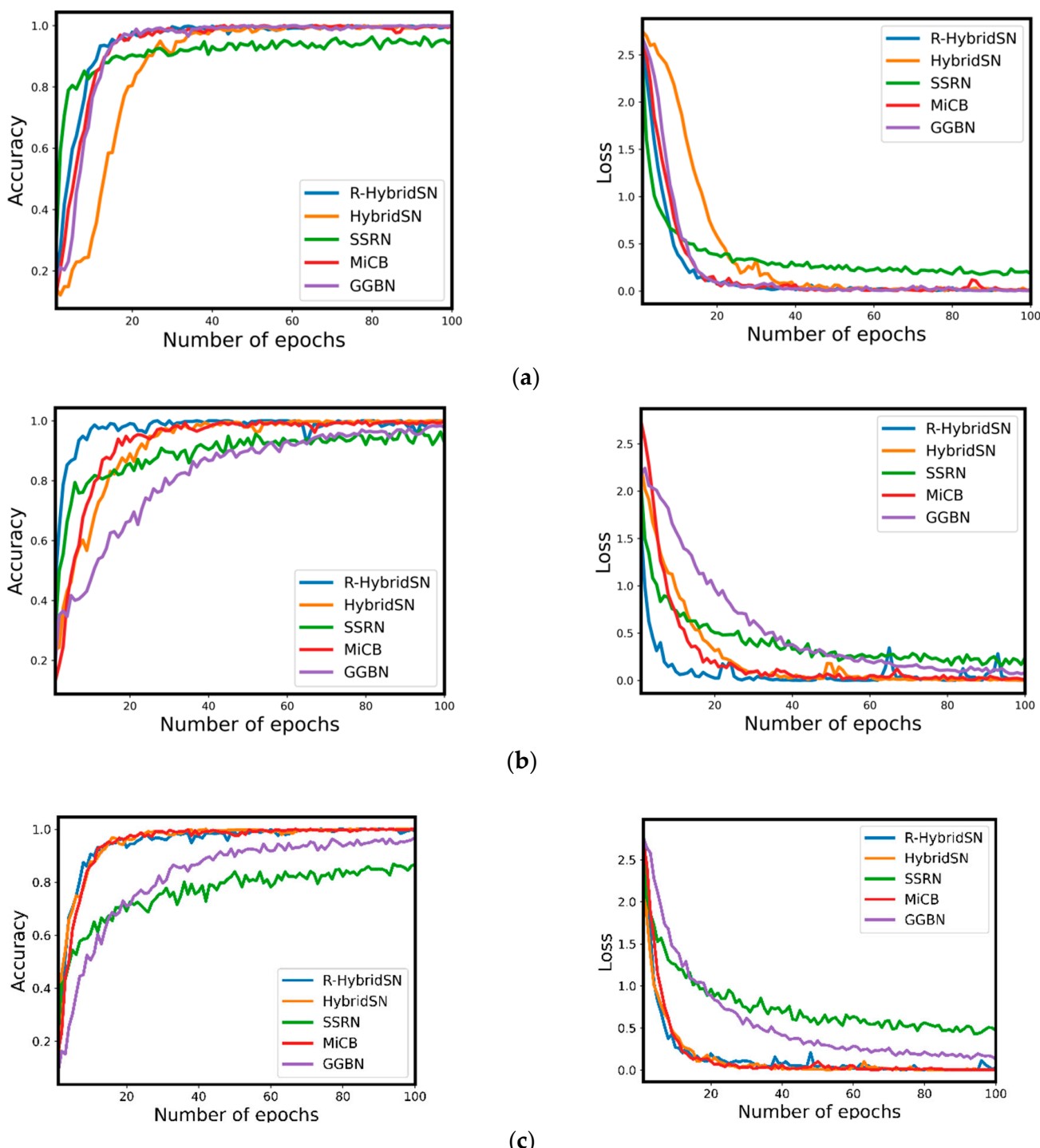

**Figure 16.** Training and loss graphs of the SSRN, HybridSN, R-HybridSN, and MiCB networks over the (**a**) IP, (**b**) UP, and (**c**) SA datasets.

*4.4. The Confusion Matrix*

In this section, we further demonstrate the competitiveness and robustness of the proposed MiCB model against the GGBN, HybridSN, and R-HybridSN over the IP, UP, and SA datasets when a small amount of training sample data are used, as shown in Figures 17–19.

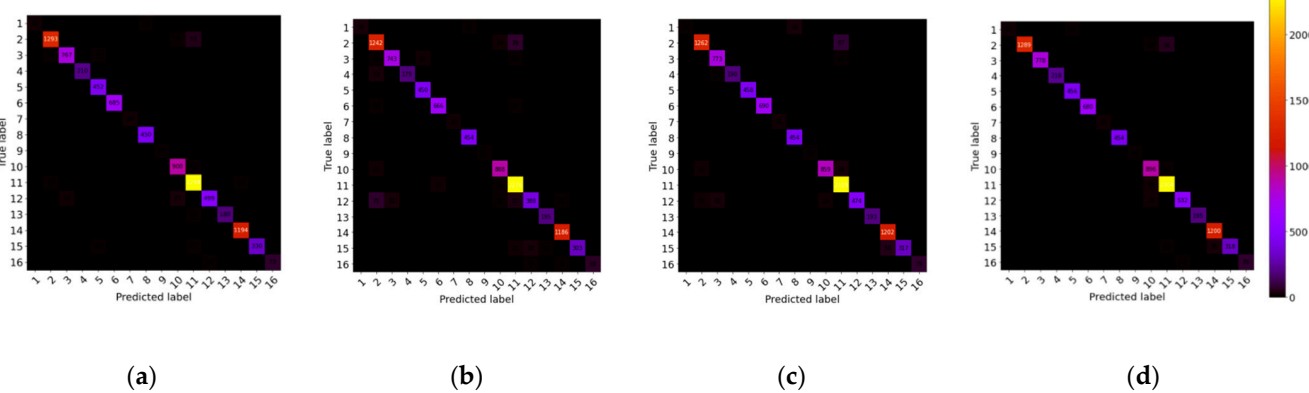

**Figure 17.** Confusion matrices over the IP dataset for (**a**) GGBN, (**b**) HybridSN, (**c**) R-HybridSN, and (**d**) MiCB.

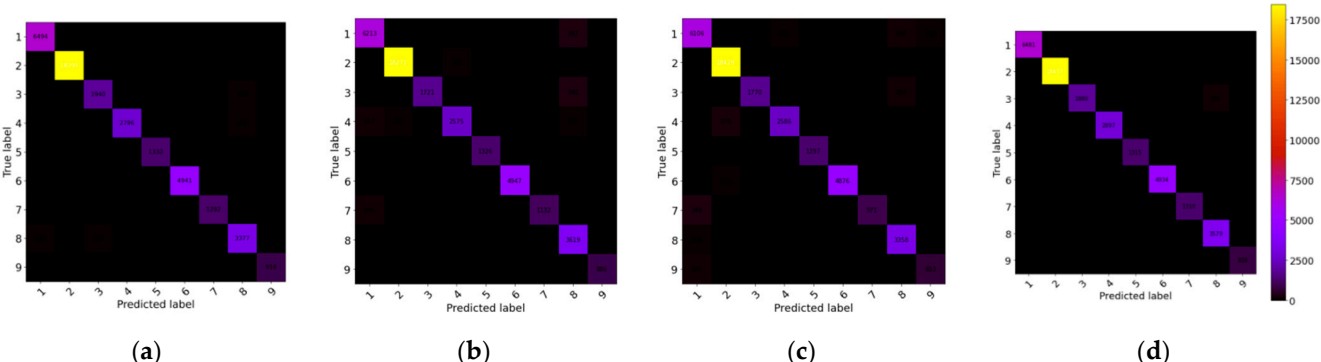

**Figure 18.** Confusion matrices over the UP dataset for (**a**) GGBN, (**b**) HybridSN, (**c**) R-HybridSN, and (**d**) MiCB.

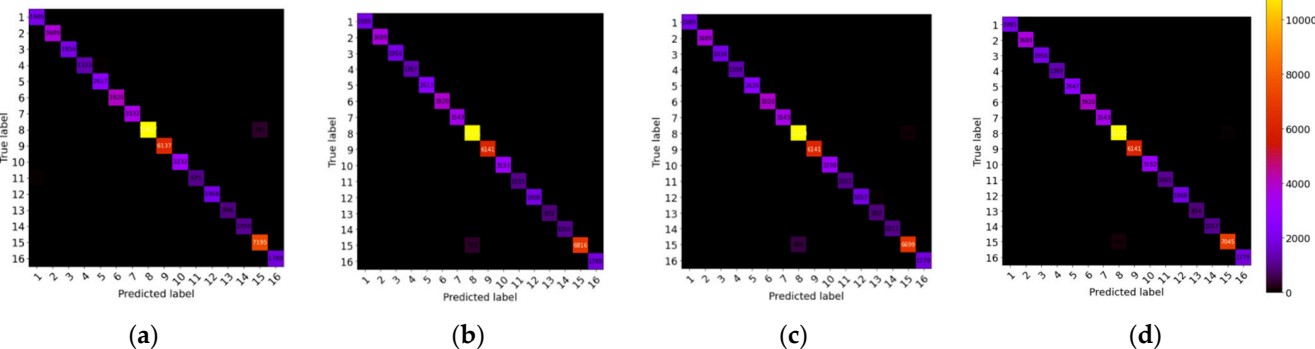

**Figure 19.** Confusion matrices over the SA dataset for (**a**) GGBN, (**b**) HybridSN, (**c**) R-HybridSN, and (**d**) MiCB.

From Figures 17–19, it can be observed that most of the sample data of the proposed MiCB model lie in a diagonal line. This demonstrates the competitiveness and robustness of the model when trained on very limited sample data.

### 4.5. Classification Diagrams

This subsection demonstrates the competitiveness and robustness of the MiCB model using the classification diagrams.

Figures 20–22 illustrate that, in contrast to the proposed MiCB model, the GGBN, HybridSN, and R-HybridSN classification maps exhibit more noisy spots over the three

benchmarking datasets. Hence, with less training sample data, the proposed MiCB model produces less noisy dispersed points and delivers smoother classification results.

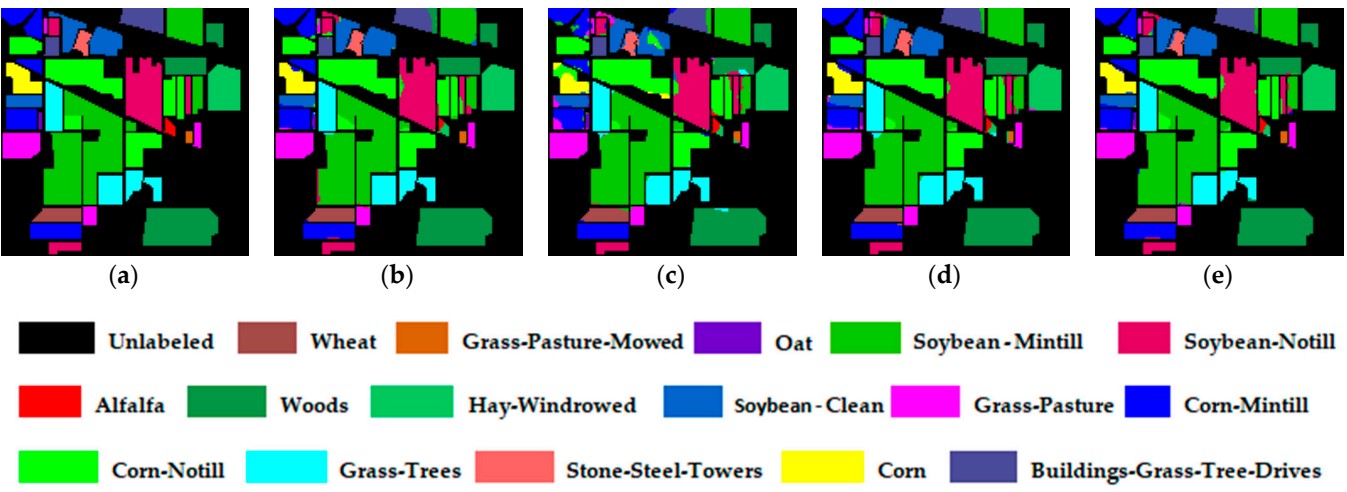

**Figure 20.** Classification maps of IP dataset: (**a**) ground truth; (**b**) R-hybridSN; (**c**) HybridSN; (**d**) GGBN; (**e**) MiCB.

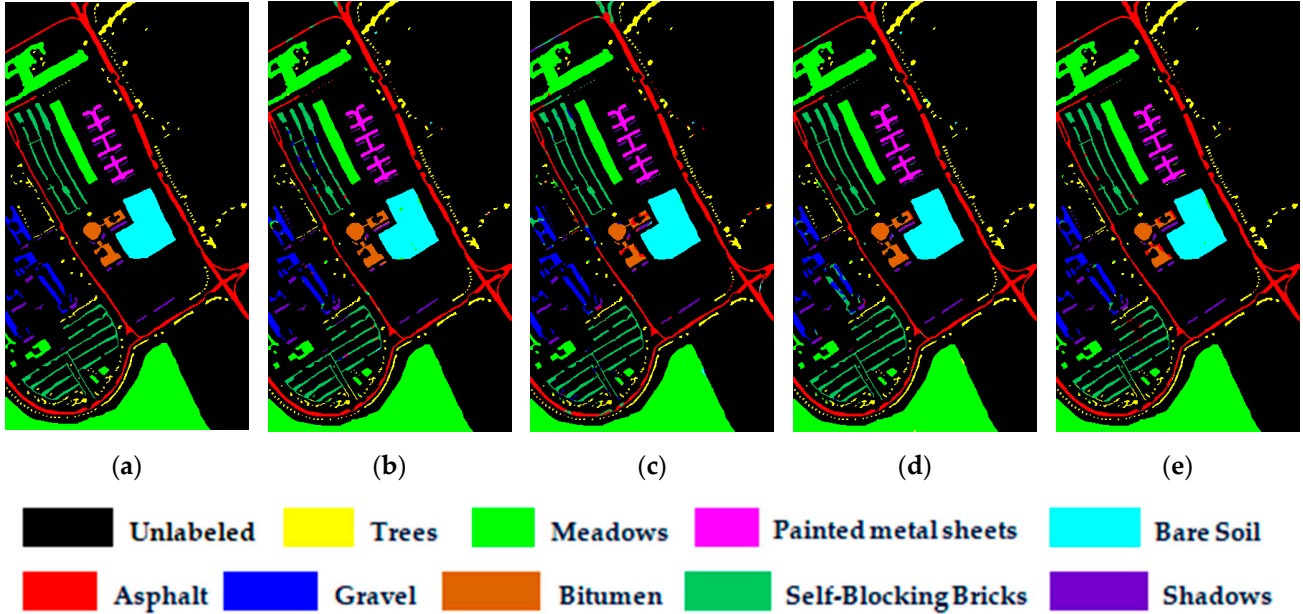

**Figure 21.** Classification maps of UP dataset: (**a**) ground truth; (**b**) R-hybridSN; (**c**) HybridSN; (**d**) GGBN; (**e**) MiCB.

*4.6. Comparison with Other Methods*

This section details the experimental results in varying the training sample data across all three datasets. Using the IP dataset, we randomly trained the models on 2%, 5%, 10%, and 20% of the total sample data. However, in UP and SA, we trained the models on 0.4%, 0.8%, 1%, 2%, and 5% of the total sample data for the UP and SA datasets and utilized the remaining sample data to test the models. The HSI classifiers performed significantly poor on the IP dataset, especially when the training sample is reduced to below 5% because some classes might be missing in the training dataset. The outcome is summarized in Tables 9–11 and Figure 23a–c.

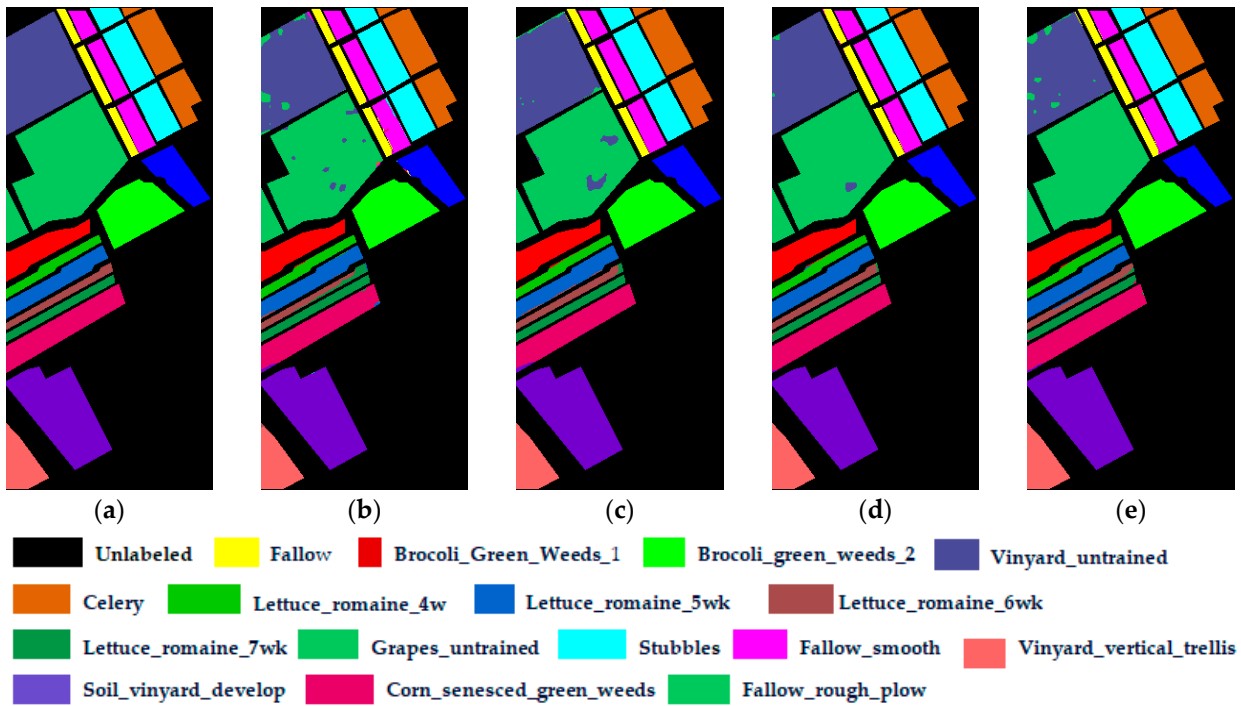

**Figure 22.** Classification maps of SA dataset: (**a**) ground truth; (**b**) R-hybridSN; (**c**) HybridSN; (**d**) GGBN; (**e**) MiCB.

**Table 9.** The influence of altering the training data samples on overall accuracy for MiCB in comparison with the selected models over the IP dataset.

| Model | \multicolumn{5}{c}{Training Sample Data in Percentage} |
| --- | --- | --- | --- | --- | --- |
| | **20%** | **10%** | **8%** | **5%** | **2%** |
| M3D-CNN | 90.03 | 80.10 | 78.04 | 68.88 | 62.28 |
| SSRN | 98.91 | 97.25 | 96.33 | 93.39 | 84.30 |
| HybridSN | 99.30 | 97.66 | 96.37 | 94.24 | 83.14 |
| R-HybridSN | 99.52 | 98.44 | 98.12 | 96.46 | 86.67 |
| GGBN | 99.45 | 98.80 | 98.04 | 96.85 | 89.37 |
| MiCB | 99.52 | 98.75 | 98.43 | 97.35 | 91.59 |

**Table 10.** The influence of altering the training data samples on overall accuracy for MiCB in comparison with the selected models over the UP dataset.

| Model | \multicolumn{5}{c}{Training Sample Data} |
| --- | --- | --- | --- | --- | --- |
| | **5%** | **2%** | **1%** | **0.80%** | **0.40%** |
| M3D-CNN | 92.80 | 89.27 | 87.19 | 82.75 | 76.53 |
| SSRN | 99.57 | 99.07 | 97.67 | 97.12 | 93.41 |
| HybridSN | 99.45 | 97.86 | 95.86 | 93.30 | 85.95 |
| R-HybridSN | 99.47 | 98.47 | 96.40 | 95.64 | 91.60 |
| GGBN | 99.74 | 99.34 | 98.13 | 97.46 | 94.66 |
| MiCB | 99.74 | 99.16 | 98.29 | 97.48 | 94.21 |

**Table 11.** The influence of altering the training data samples on overall accuracy for MiCB in comparison with the selected models over the SA dataset.

| Model | Training Sample Data | | | | |
|---|---|---|---|---|---|
| | **5%** | **2%** | **1%** | **0.80%** | **0.40%** |
| M3D-CNN | 92.65 | 90.17 | 88.02 | 86.82 | 83.42 |
| SSRN | 98.7 | 98.02 | 96.94 | 96.87 | 93.64 |
| HybridSN | 99.83 | 99.57 | 98.72 | 97.78 | 94.88 |
| R-HybridSN | 99.82 | 99.36 | 98.25 | 96.97 | 94.33 |
| GGBN | 99.97 | 99.68 | 99.29 | 98.32 | 97.26 |
| MiCB | 99.93 | 99.73 | 99.20 | 98.57 | 96.07 |

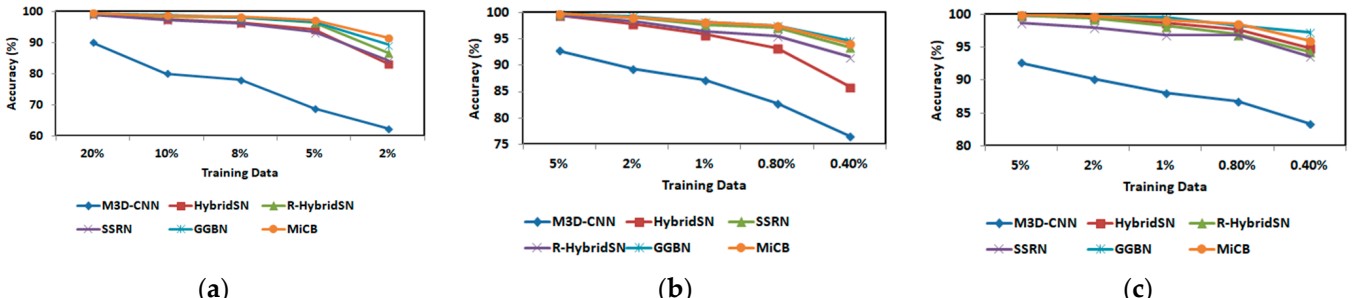

(**a**)                (**b**)                (**c**)

**Figure 23.** The effect of varying the number of training samples for the M3D-CNN, HybridSN, R-HybridSN, SSRN, GGBN, and MiCB methods on the overall accuracy (OA) over (**a**) IP; (**b**) UP; (**c**) SA datasets.

The increase in training sample data narrows the OA gap between the compared methods and converges at some point, as illustrated in Tables 9–11 and Figure 23a–c. For example, in the IP dataset, when 5% of the data are used to train the models, the MiCB increases the OA gap of M3D-CNN, HybridSN, R-HybridSN, SSRN, GGBN by +28.47%, 3.11%, +0.89%, +3.96%, and +0.5%, respectively. When the training sample data are 20% of the total data, the MiCB increases the OA gap of M3D-CNN, HybridSN, R-HybridSN, SSRN, and GGBN by +9.49%, +0.22%, 0.0%, +0.61%, and +0.07%, respectively. This observation is clearly visualized in Table 9 and Figure 23a. In addition, we note that when the comparison models are trained on extremely small sample sizes, the MiCB model achieves the highest classification accuracy in almost all the datasets, thus showing the model's adaptability. For instance, the MiCB improved the OA of M3D-CNN, HybridSN, R-HybridSN, SSRN, and GGBN by +29.31%, +8.45%, +4.92%, +7.29%, and +2.22% on 2% IP train data (see Table 9); by +14.73%, +4.18%, +0.36%, +1.84%, +0.02% on 0.40% UP train data (see Table 10 and Figure 23b); and by +11.75%, +0.79%, +1.6%, +1.7%, and +0.25% on 0.8% SA train data (see Table 11 and Figure 23c), respectively.

## 5. Conclusions

This paper aims to add to the scientific work of making deep networks for HSI classification that can optimally train with merger training samples while mitigating the overfitting problem. This paper proposes an integrated deep multi-scale 3D/2D convolutional network block (MiCB) for simultaneous low-level spectral and high-level spatial feature extraction that can optimally be trained with a limited amount of training sample data. The primary contribution of the MiCB model is its creative use of MiCB blocks, which allow the network to convolve low-level spectral features with high-level spatial features that are strengthened by multi-scale kernels, residual connections, and depthwise separable convolutions. The use of non-identity multi-residual connections in the MiCB network drastically reduces the challenge of gradient disappearance in the MiCB network. Exploding network parameters can be addressed by replacing the traditional 2D-CNN with the 2D depthwise separable convolutional layers, which also prevents overfitting as the model structure deepens. Lastly,

utilizing multi-scale kernels promotes the extraction of highly discriminative features and increases the generalizability of the model. The innovative combination of these four approaches in the MiCB network structure enables the model to extract distinct and abundant contextual features to achieve high classification accuracy even with few training samples. We tested the robustness and competitiveness of our model with cutting-edge methods over the IP, UP, and SA datasets. Our proposed method achieves better classification accuracy than M3D-CNN, HybridSN, SSRN, R-HybridSN, and comparable results with GGBN.

**Author Contributions:** Conceptualization, H.C.T., E.C. and D.O.N.; software, H.C.T. and D.O.N.; resources, E.C.; writing—original draft preparation, H.C.T. and D.O.N.; writing—review and editing, H.C.T., E.C. and D.O.N.; supervision, E.C.; funding acquisition, E.C. All authors have read and agreed to the published version of the manuscript.

**Funding:** This work was sponsored in part by grants 62101503 and 62101505 from the National Natural Science Foundation of China, and Henan Science and Technology Research Project under grant 222102210102.

**Data Availability Statement:** All datasets used in this research are openly accessible online (http://www.ehu.eus/ccwintco/index.php?title=Hyperspectral_Remote_Sensing_Scenes, accessed date on 20 June 2023).

**Conflicts of Interest:** There are no conflicts of interest declared by the authors.

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
