# Peer review of "Improving Feature Learning in Remote Sensing Images Using an Integrated Deep Multi-Scale 3D/2D Convolutional Network"

_remotesensing, doi:10.3390/rs15133270_

Round 1

Reviewer 1 Report

The network structure in this manuscript only combines existing network models, and the classification process of this approach is general. In experiment section, three hyperspectral datasets are acquired very early, and their image sizes are very limited. Using only these three datasets is not very convincing, and classification accuracies of the proposed method on these datasets do not have significant advantages.

Fine

Author Response

Thank you for providing us with an opportunity to address the comments you raised. Kindly find our point-by-point response to the comments (below), and an updated manuscript with changes.

Reviewer Comment:

In experiment section, three hyperspectral datasets are acquired very early, and their image sizes are very limited.  Using only these three datasets is not very convincing, and classification accuracies of the proposed method on these datasets do not have significant advantages.

Author Response:

Most of the freely available datasets were produced in almost the same time as IP, SA and UP. The new dataset is Houston Dataset 2013 and 2018, which requires permission before we can download. We wrote to them, however, we have not received any feedback. Given the short deadline to submit the response, we kindly request you to allow us submit our experiment using the existing IP, UP and SA datasets. Moreover, the compared methods only used the three famous datasets: IP, SA and UP. However, in future experiments, we shall endeavor to include the latest datasets available.  

Best regards,

Tinega, Chen, and Nyasaka

Reviewer 2 Report

This paper proposes an integrated deep multi-scale 3D/2D convolutional network. The structure of the manuscript is not well organized. The contribution parts of this manuscript content need to be further clearly improved. In general, many obvious problems exist. The reviewer suggest that the paper can only be considered for possible publication after a thoroughly major revision.

The detailed comments are as follows:

1) The dataset IP, UP and SA is too old. Please compare all of methods on new datasets.

2) Line 175 Page 7, the number of ‘figure 10’ is chaotic.

3) Once each channel images are stacked and combined to make one big 2D image, the correlation information between neighbor channels is collapsed. This may reduce the merit from the 3D convolution processes.

4) All the comparison methods are old (HybridSN 2017, SSRN 2018). You must increase the comparison algorithms including 3D- and 2D-based convolutional methods. Conclusion is not clear.

5) Please provide the training and testing time of compared algorithms.

6) The performance of classification in Figure 22 (c) is better than Figure 22 (e).

  • There are some presentation problems, I suggest careful checking

Author Response

Thank you for providing us with an opportunity to address the comments you raised. Kindly find our point-by-point response to the comments (below), and an updated manuscript with changes.  

Reviewer Comment 1:

The dataset IP, UP and SA is too old. Please compare all of methods on new datasets.

Author Response:

Most of the freely available datasets were produced in almost the same time as IP, SA and UP. The new dataset is Houston Dataset 2013 and 2018, which requires permission before we can download. We wrote to them, however, we have not received any feedback. Given the short deadline to submit the response, we kindly request you to allow us submit our experiment using the existing IP, UP and SA datasets. Moreover, the compared methods only used the three famous datasets: IP, SA and UP. However, in future experiments, we shall endeavor to include the latest datasets available.

Reviewer Comment 2:

Line 175 Page 7, the number of ‘figure 10’ is chaotic.

Author Response:

We kindly request you advise us what we can do with the diagram.

Reviewer Comment 3:

All the comparison methods are old (HybridSN 2017, SSRN 2018). You must increase the comparison algorithms including 3D- and 2D-based convolutional methods. Conclusion is not clear.

Author Response:

Thank you for noting this observation. It is true that some of the methods used such as Hybrid SN and SSRN are old. This is because, this experiment is a continuation of previous experiments that used these models and also, we have tried to include R-HybridSN which was published in November 2019.

Reviewer Comment 4:

Please provide the training and testing time of compared algorithms.

Author Response:

We welcome this suggestion. We have included the training and testing time of the compared algorithms as advised.

Reviewer Comment 5:

The performance of classification in Figure 22 (c) is better than Figure 22 (e).

Author Response:

Thank you for noting this concern; we have corrected it by rearranging the images.

Reviewer Comment 6:

There are some presentation problems, I suggest careful checking

Author Response:

Thank you for noting this concern, we have improved the presentation of the paper to the best of our knowledge.

Best regards,

Tinega, Chen, Nyasaka

Reviewer 3 Report

The manuscript presents an integrated deep multi-scale 3D/2D convolutional network (MiCB) for simultaneous low-level spectral and high-level spatial features extraction in hyperspectral images (HSI) that can be optimally trained using limited training samples. The presented experimental results demonstrate the efficacy of the proposed MiCB model when small amounts of training sample data are considered (less than 5% of the samples) since the obtained classification results are better than alternative state-of-the-art methods.

From the technical point of view, I find that the manuscript addresses an interesting topic that matches the scope of the Remote Sensing MDPI open-access journal and that the presented work is of good quality. Nevertheless, I believe that there are a couple of aspects that the authors must revise before the manuscript is accepted for publication, namely:

1. The state of the art review presented in section I only lists (or very succinctly describes) relevant related works. A discussion about the limitations of such approaches is lacking. Also, the authors do not clearly state which of those limitations they propose to overcome with this work.

2. Formulas (1), (2), and (3) and the companion texts should be reviewed.

3. In section 2.4, the authors should explain better the advantages of the depth-separable convolution for HSI classification.

4. An alternative color scheme should be used for Figure 17 so that the plots are readable.

5. It is quite difficult (almost impossible) to draw the conclusions presented in section 4.6 from figures 21-23. This must be reviewed.

In what concerns the presentation, I find that the submitted manuscript is well written and properly structured, thus being easy to read. Nevertheless, there are some aspects that can be improved. For example, almost all the tables and figures are misplaced in the manuscript. Also, several references must be reviewed to correct style issues. More specific comments can be found in the PDF file attached to my review form.

In conclusion, IMO, the manuscript addresses an interesting topic but there are some minor aspects that must be improved before the manuscript is accepted for publication.

The English language is fine but there are still some typos that must be corrected.

Author Response

Thank you for providing us with an opportunity to address the comments you raised. Kindly find our point-by-point response to the comments (below), and an updated manuscript with changes.  

Reviewer Comment 1:

The state-of-the-art review presented in section I only lists (or very succinctly describes) relevant related works. A discussion about the limitations of such approaches is lacking. Also, the authors do not clearly state which of those limitations they propose to overcome with this work.

Author Response:

We appreciate this comment. In section I, we presented three categories of deep CNN feature learning methods based on how HSI features are processed: pre-processing-based, post-processing-based, and integrated-based. We have included limitations at the end of each category as shown in the edited manuscript.

Reviewer Comment 2:

Formulas (1), (2), and (3) and the companion texts should be reviewed.

Author Response:

Thank for noting the mistake, we have edited appropriately as shown in the edited manuscript.

Reviewer Comment 3:

In section 2.4, the authors should explain better the advantages of the depth-separable convolution for HSI classification.

Author Response:

Thank you for noting this comment. We have explained the advantages of the depth-separable convolution for HSI classification to the best of our advantage. Kindly check the updated manuscript.

Reviewer Comment 4:

An alternative color scheme should be used for Figure 17 so that the plots are readable.

Author Response:

Thank you for noting this comment. However, we are using the heatmap in the color map to plot the confusion matrix in order to show the dispersion of values. We opted for this option because the use of values alone might be too small to be readable when representing confusion matrix with wide range of confusion values as observed in this paper, and when presenting the confusion matrices in the format presented in the paper.

Reviewer Comment 5:

Nevertheless, there are some aspects that can be improved. For example, almost all the tables and figures are misplaced in the manuscript.

Author Response:

Thank you for noting this mistake. We have reviewed for all the tables and figures in the manuscript and edited as advised.

Reviewer Comment 6:

Also, several references must be reviewed to correct style issues. More specific comments can be found in the PDF file attached to my review form.

Author Response:

We have reviewed the paper as per the reference. We thank you for the through job.

Best regards,

Tinega, Chen, Nyasaka

Round 2

Reviewer 1 Report

Considering the limited revising time and the accessibility of datasets, the current manuscript seems better than the original one and I could recommend this manuscript for publication in Remote Sensing now.

If the manuscript still needs some further revision, I still suggest the authors add more experiments conducted on newer datasets. 

Author Response

Thank you for providing us with an opportunity to address the comments you raised.

Reviewer suggestion:

If the manuscript still needs some further revision, I still suggest the authors add more experiments conducted on newer datasets.

Author Response

We have added one new model to the experiment. As for adding a new dataset, we contacted the owner of Houston 2013 and 2018 dataset, but he has not responded. We hope in the future to use the dataset.  Kindly find the attached manuscript for verification

Reviewer 2 Report

Although it is a continuation of the previous method, I personally think it should be compared with the new method.

Although it is a continuation of the previous method, I personally think it should be compared with the new method.

Author Response

Thank you for providing us with an opportunity to address the comments you raised. 

Reviewer Comment

Although it is a continuation of the previous method, I personally think it should be compared with the new method

Author Response

Thank you for this suggestion. We have included the GGBN a model that was published in 2021. Kindly, find the attached updated manuscript on the same
